# Identification of a Novel Anti-HIV-1 Protein from *Momordica balsamina* Leaf Extract

**DOI:** 10.3390/ijerph192215227

**Published:** 2022-11-18

**Authors:** Morgan I. Coleman, Mahfuz Khan, Erick Gbodossou, Amad Diop, Kenya DeBarros, Hao Duong, Vincent C. Bond, Virginia Floyd, Kofi Kondwani, Valerie Montgomery Rice, Francois Villinger, Michael D. Powell

**Affiliations:** 1Department of Microbiology, Biochemistry and Immunology, Morehouse School of Medicine, 720 Westview Dr. SW, Atlanta, GA 30310, USA; 2PROMETRA International, Dakar-Etoile BP 6134, Senegal; 3Malango Traditional Healers Association, Fatick BP 1763, Senegal; 4Department of Pharmacology and Toxicology, Morehouse School of Medicine, 720 Westview Dr. SW, Atlanta, GA 30310, USA; 5Department of Community Health and Prevention, Morehouse School of Medicine, 720 Westview Dr. SW, Atlanta, GA 30310, USA; 6Office of the President, Morehouse School of Medicine, 720 Westview Dr. SW, Atlanta, GA 30310, USA; 7Department of Biology Director, New Iberia Research Center, University of Louisiana at Lafayette, 4401 W Admiral Doyle Drive, New Iberia, LA 70560, USA

**Keywords:** HIV-1, MoMo30, fusion inhibitor, extract, antiviral, antiviral protein, antiviral plant

## Abstract

Our lab investigates the anti-HIV-1 activity in *Momordica balsamina* (*M. balsamina*) leaf extract. Traditional Senegalese healers have used *M. balsamina* leaf extract as a part of a plant-based treatment for HIV/AIDS infections. Our overall goal is to define and validate the scientific basis for using *M. balsamina* leaf extract as a part of the traditional Senegalese treatment. As an initial characterization of this extract, we used activity-guided fractionation to determine the active ingredient’s solubility and relative size. We found that *M. balsamina* leaf extract inhibits HIV-1 infection by >50% at concentrations of 0.02 mg/mL and above and is not toxic over its inhibitory range (0–0.5 mg/mL). We observed significantly more antiviral activity in direct water and acetonitrile extractions (*p* ≤ 0.05). We also observed significantly more antiviral activity in the aqueous phases of ethyl acetate, chloroform, and diethyl ether extractions (*p* ≤ 0.05). Though most of the antiviral activity partitioned into the aqueous layers, some antiviral activity was present in the organic layers. We show that the active agent in the plant extracts is at least 30 kD in size. Significantly more antiviral activity was retained in 3, 10, and 30 kD molecular weight cutoff filters (*p* ≤ 0.05). In contrast, most of the antiviral activity passed through the 100 kD filter (*p* ≤ 0.05). Because the active anti-HIV-1 agent presented as a large, amphiphilic molecule we ran the purified extract on an SDS-page gel. We show that the anti-HIV-1 activity in the leaf extracts is attributed to a 30 kDa protein we call MoMo30. This article describes how MoMo30 was determined to be responsible for its anti-HIV-1 activity.

## 1. Introduction

Natural products are an essential source of biologically active molecules. Nearly half of the drugs approved by the United States Food and Drug Administration (F.D.A.) over the past 40 years have used natural products [1]. With the increased incidence of drug-resistant pathogens, researchers are turning more and more to traditional folk remedies as viable treatment options. One such disease is the human immunodeficiency virus/acquired immunodeficiency virus (HIV/AIDS). The depletion of CD4+T cells characterizes HIV/AIDS. Lack of immune cells weakens the body’s immune response and increases susceptibility to opportunistic pathogens [2]. Current therapies for HIV-1 involve lifelong use of a combination of antiretroviral therapies (cART). Although therapies for HIV/AIDS have improved over the years, drawbacks still heavily influence the patient’s quality of life, especially long-term [3]. These drawbacks include drug toxicity, drug resistance, and accessibility to treatments. To that end, our lab is exploring a traditional Senegalese remedy to treat HIV-1 infection through a partnership with an organization called PROMETRA International. We are testing a plant-based anti-HIV-1 therapy used by traditional Senegalese healers. One plant used in this therapy is *Momordica balsamina*. The treatment is brewed as a tea and taken orally.

*Momordica balsamina* (*M. balsamina*), commonly known as balsam apple and African pumpkin, is a tendril-bearing plant from the Cucurbitaceae family, native to the tropical climates of Africa and Southeast Asia [4,5]. There is considerable research surrounding the anticancer, antiviral, antifungal, antiparasitic, and antibacterial properties associated with *M. balsamina.* For example, *M. balsamina* is considered a great source of ascorbic acid and antioxidants and is a part of the daily diet of many people. It is also promoted as a protein supplement in communities of low socioeconomic status. The anti-inflammatory properties have also been studied. *M. balsamina* is used to treat ulcers, reduce blood pressure, and treat wounds [4,5]. Most notably, there have been numerous reports on antiviral proteins found within the seed of *M. balsamina*. The antiviral activity in the seed extract has been attributed to various small molecules and antiviral proteins. These molecules work by irreversibly inhibiting viral translation, thus inhibiting the propagation of infection [6,7,8,9]. Because *M. balsamina* has been established as an antiviral plant, it is important to consider the traditional Senegalese treatment as a viable treatment option.

This study aims to determine if *M. balsamina* leaf extracts have anti-HIV-1 activity and the nature of the active ingredient(s) responsible for this activity. To make these determinations, we have used activity-guided fractionation to determine the active agents’ relative solubility and size [10]. We will use this data to form the basis for further investigation into the detailed mechanism of action of this traditional treatment for HIV/AIDS.

## 2. Materials and Methods

### 2.1. Processing of Plant Materials

*Momordica balsamina* (*M. balsamina*) was grown and harvested in Dakar-Etoile, Senegal, by our partners at PROMETRA International. Botanists at PROMETRA identified and collected fresh *M. balsamina* leaves. PROMETRA cleaned the *M. balsamina* leaves to remove any particulates and removed stems and prior to drying them at room in the open air in the sun for not more than five days. The dried leaves were crushed using a plant mill (Everson, EWF-20) then added to water. The mixture was boiled at 100 °C for 30 min and air-dried into a powder before shipping.

### 2.2. Preparation of Stock Solution for Dose-Reponse and Toxicity Testing

After the plant materials arrived at our lab, 100 g of powder was dissolved in 1 L of sterile distilled water at 4 °C overnight. It was then extensively filtered using a Whatman #3 filter paper and centrifuged (Eppendorf 5810R, Hamburg, Germany) at 4000× *g* for 30 min to remove potential particulate contamination. We sterilized the extract using a 0.45-micron filter and then lyophilized (Thermo Savant MODULYOD-115, Waltham, MA, USA) it into a powder. We prepared a 10 mg/mL stock solution of leaf extract in ultrapure water.

### 2.3. Dose–Response Curve

The stock solution was diluted with ultrapure water to make the following concentrations: 0.02 mg/mL, 0.05 mg/mL, 0.1 mg/mL, 0.2 mg/mL, 0.5 mg/mL, and 1 mg/mL. The dilutions were used in MAGI infectivity assays to determine anti-HIV-1 activity at each concentration. The baseline for infectivity was the untreated cells (0 mg/mL). Infectivity assays were performed in triplicate for each concentration.

### 2.4. Cytotoxic (MT.) Assay

The stock solution (20 mg/mL stock solution). The stock solution was diluted with ultrapure water to make the following concentrations: 0.02 mg/mL, 0.05 mg/mL, 0.1 mg/mL, 0.2 mg/mL, 0.5 mg/mL, and 1 mg/mL. The dilutions were used in 3-(4,5-dimethylthiazol-2-yl)-2,5-diphenyl-2H-tetrazolium bromide (MTT) assays. We used the Promega M.T.T. Assay kit to determine cell viability. We conducted the M.T.T. assay suggested by the manufacturer. The cell media did not contain phenol red, as phenol red interferes with some M.T.T. assays. Each assay was performed in triplicate.

### 2.5. Solvent Extractions

#### 2.5.1. Direction Extractions (Water, Isopropyl Alcohol, Tetrahydrofuran, Acetonitrile, Ethanol, Acetone)

Avomeen Analytical Chemistry Services prepared direct extractions. Essentially, 5–7 mg of the dried *M. balsamina* powder was dissolved in 10.0 mL of the following solvents: water, isopropyl alcohol, tetrahydrofuran, acetonitrile, ethanol, and acetone. The solution was vortexed for 3–5 min, then centrifuged at 4000× *g* for 15 min. Each supernatant was used as a treatment in separate MAGI infectivity assays to assess viral inhibition.

#### 2.5.2. Phase Extractions (Ethyl Acetate, Chloroform, Hexane, Diethyl Ether)

Avomeen Analytical Chemistry Servies prepared the phase extractions. In essence, 5–7 mg of dried *M. balsamina* powder was dissolved in a water solution and mixed with one of the following solvents: ethyl acetate, chloroform, hexane, and diethyl ether. The solutions were vortexed for 3–5 min. The mixture was centrifuged, and phase separation occurred. 1.0 mL of the organic phase and 1.0 mL of the aqueous phase were collected. The organic phase extract was dried in a turbo-vac at room temperature to remove the organic solvent and reconstituted in an aqueous solution for testing. The collected extractions from the aqueous and organic phases were used as treatments in separate MAGI infectivity assays to assess viral inhibition.

### 2.6. Characterization Using Molecular Weight Cutoff

We prepared a 2 mg/mL stock solution of *M. balsamina* leaf extract. We passed the extract through Amicon filters of increasing pore size: 3 kD, 10 kD, 30 kD, and 100 kD molecular weight cutoff filters. 1.0 mL of the filtrate and retained solutions were collected and used in separate MAGI infectivity assays.

### 2.7. MAGI Infectivity Assays

Viral inhibition was determined using MAGI infectivity assays [11,12]. We adapted the protocol used in the cited literature.

#### 2.7.1. Maintenance of Cells Prior to Infection/Treatment

HeLa-CD4+-LTR-βgal cells (ATCC Cat# ARP-3596) were maintained in Dulbecco-modified eagle medium (DMEM). Prior to the infectivity assay, the cells were supplemented with fetal bovine serum (FBS) (5%), G418 (0.1 mg/mL), hygromycin B (0.05 mg/mL), L-glutamine, penicillin (100 μg/mL), and streptomycin (100 μg/mL). We seeded 2 × 10^5^ cells/well in six-well plates 24 h before the infectivity assay.

#### 2.7.2. Preparation of Treatment Solution for Infectivity Assays

We prepared the treatment solutions in 15 mL tubes. Each treatment solution consisted of the following:1 mL extract (prepared in Section 2.3 and Section 2.5)1 ng HIV-1_NL4-3_2 µL of diethylaminoethyl-dextran (DEAE) (200 µg/mL stock solution)DMEM with 5% F.B.S. to a total volume of 2 mL

#### 2.7.3. Addition of Treatment Solution to Cells

We removed the maintenance media from the HeLa-CD4+-LTR-βgal cells and added 2 mL of the testing solution per well. The cells were incubated at 37 °C in 5% CO_2_ for 2 h. Afterward, we removed the testing solutions from the wells and added 3 mL of DMEM with 5% F.B.S. We then incubated the cells for 48 h at 37 °C in 5% CO_2_. Extract and virus were added to the cells simultaneously because preliminary data suggests that pre-drugging the cells does not affect viral inhibition.

#### 2.7.4. Fixing and Staining Cells

After the 48 h incubation, we removed the media from the cells and fixed them in a phosphate-buffered saline (PBS) solution containing 1% formaldehyde and 0.2% glutaraldehyde for approximately 3–5 min. Next, the cells were washed with PBS twice. Then, we stained the cells in a PBS solution containing X-gal solution (5-bromo-4-chloro-3-indolyl-β-D-galactosidase) dissolved in DMSO (40 mg/mL), 4 mM potassium ferrocyanide, 4 mM potassium ferricyanide, and 2 mM magnesium chloride. The cells remained in the staining solution for 50 min at 37 °C. Next, we washed the cell two times with PBS. After the last wash, we added PBS and counted the number of infected cells in the well. This experiment’s control consisted of HeLa-CD4+-LTR-βgal cells with no leaf extract added, HeLa-CD4+-LTR-βgal cells with no NL4-3 added, and HeLa-CD4+-LTR-βgal cells with only media (no leaf extract or NL4-3). Each assay was performed in triplicate. We calculated the percentage of viral inhibition using the following equation:$$Percent\;Viral\;Inhibition=\left[1-\left(\frac{Number\;of\;infected\;cells\;in\;sample}{Number\;of\;infected\;cells\;in\;virus\;only\;control}\right)\right]\times 100\%$$

To ensure that the active agent in the leaf extract does not inhibit β-gal activity, we used a β-gal assay kit (Promega Cat# E2000) to measure β-gal activity in the presence of extract. We completed the protocol as suggested by the manufacturer. Water was used as a negative control.

### 2.8. Protein Gel and Coomassie Brilliant Blue Stain

The water-soluble extracts were run on an SDS-PAGE (4–20%) protein gel for 30 min at 200 V. The gel was then stained in Coomassie brilliant blue (C.B.B.) stain (Bio-Rad Coomassie Brilliant Blue R-250 Cat# 161-0400) for 1 h at room temperature, with gentle shaking.

## 3. Results

### 3.1. Dose–Response Curve and Cytotoxic Assay

To determine if water-soluble extracts of *M. balsamina* contained anti-HIV-1 activity, we infected HeLa-CD4+-LTR-βgal cells with HIV-1_NL4-3_ and treated them with varying concentrations (0–1 mg/mL) of leaf extract. The water-soluble leaf extract inhibited HIV-1_NL4-3_ infection at all doses. Approximately 1200 cells were infected at baseline. The 0.02 mg/mL dose of leaf extract decreased the number of infected cells by 75%. The 1 mg/mL dose of leaf extract completely inhibited HIV-1_NL4-3_ infection. We summarize the results in Figure 1A.

Cytotoxicity of *M. balsamina* leaf extract was determined using the 3-(4,5-dimethylthiazol-2-yl)-2,5-diphenyl-2H-tetrazolium bromide (MTT). We ran a one-way ANOVA to compare the cytotoxicity of the untreated cells (0 mg/mL) with the different concentrations of leaf extract. The leaf extract was not toxic at inhibitory levels ranging from 0.02 mg/mL to 0.5 mg/mL. Cells treated with 0.02–0.5 mg/mL of leaf extract remained stable, with no significant changes in cytotoxicity. However, cytotoxicity increased significantly in the cells treated with 1 mg/mL of leaf extract. The 5 mg/mL and 10 mg/mL doses of leaf extract killed the cells. We show the results for the MTT assay in Figure 1B. We show the p-values for the untreated cells compared with each concentration of leaf extract in Table 1.

To ensure that the extract did not inhibit β-gal activity during the MAGI infectivity assay, we tested β-gal activity in water and *M. balsamina* leaf extract. The stock concentration of leaf extract was 100 µg/µL. We show the results in Table 2. There is no significant difference in β-gal activity in the presence of *M. balsamina* leaf extract. *M. balsamina* leaf extract does not influence β-gal activity.

### 3.2. Viral Inhibition of Direct Solvent Extractions

Our original consideration was that the active ingredient was likely to be a small molecule therefore we wanted information on its solubility. To determine the best solubility conditions for the antiviral ingredient within the leaf extract, we tested direct solubility in different solvents: water, isopropyl alcohol, tetrahydrofuran, acetonitrile, ethanol, and acetone. We summarize the results in Figure 2. The active ingredient was most soluble in water and acetonitrile. The active ingredient was less soluble in isopropyl alcohol, tetrahydrofuran, ethanol, and acetone. We ran a one-way ANOVA to compare the percentage of viral inhibition with each other. We found that water-soluble extract exhibited significantly more anti-HIV-1 activity than isopropyl alcohol, tetrahydrofuran, ethanol, and acetone. We show the p-values for these comparisons in Table 3.

Though significantly more anti-HIV-1_NL4-3_ activity precipitated into the most polar solvent extractions (water and acetonitrile), we observed viral inhibition in the less polar solvents. The active anti-HIV-1 agent has amphiphilic properties.

### 3.3. Viral Inhibition of Phase Extractions

Figure 3 compares the relative partitioning of the anti-HIV-1_NL4-3_ activity into each solvent’s organic and aqueous layers. Significantly more anti-HIV-1_NL4-3_ activity was partitioned into the aqueous phase of the ethyl acetate, chloroform, and diethyl ether extractions. The partition of anti-HIV-1_NL4-3_ was equivocal for the extraction in hexane. We used a two-way ANOVA to assess the results. We show the p-values in Table 4.

### 3.4. Size Approximation Using Molecular Weight Cutoff Filters

We used molecular weight cutoff filters to determine the relative size of the active ingredient. We show the percentages of viral inhibition detected in the filtrate and retained liquids in Figure 4. The filtrate (which passes through the filter) is represented with a solid bar, and the retained portion is presented with a dashed bar. Anti-HIV-1_NL4-3_ activity in the *M. balsamina* leaf extract was retained by the 3 kD, 10 kD, and 30 kD filters. In contrast, significantly more anti-HIV-1_NL4-3_ activity passed through the 100 kD filter. This is an indication that the active ingredient is at least 30 kD.

### 3.5. S.D.S.-PAGE Gel (4–20%) and Coomassie Brilliant Blue Stain

Given that the solvent studies were variable (i.e., the active agent showed both hydrophilic and hydrophobic characteristics) and the approximate size (>30 kD), we considered that the active ingredient in the extracts could be a protein. We ran the purified leaf extract on a 4–20% SDS-PAGE gel stained with Coomassie Blue (C.B.B.) to detect the presence of any proteins. Only one band at approximately 30 kD appeared. We show the stained gel with molecular weight markers in Figure 5. We also ran a gel on the retained and filtered liquids of the leaf extracts passed through the Amicon filters. We show the gel in Figure 6. An ~30 kD protein was retained in the fraction with the most anti-HIV-1_NL4-3_ activity.

## 4. Discussion

In this study, we report on the characterization of the anti-HIV-1 activity of *M. balsamina* leaf extract. We attribute this activity to a 30 kD protein we call MoMo30. Other molecules have been linked to antiviral, antibacterial, anticancer, antiparasitic, and analgesic activity in *M. balsamina* [4,5,6,7,8,9,13,14,15]. A group of 30 kD anti-HIV proteins, known as Type-1 ribosome-inactivating proteins (R.I.P.s), have previously been extracted from *M. balsamina*. R.I.P.s are a type of N-glycosidase that specifically and irreversibly inhibit protein translation” and can inhibit viral replication [6]. Although similar in size, several properties of R.I.Ps distinguish them from MoMo30. First, type-1 R.I.Ps, are found predominantly in the seeds of *M. balsamina* [7] while MoMo30 is present predominantly in leaves. Second, the N-terminal sequences are not similar. Using Edman degradation (Creative Proteomics), we were able to identify the N-terminal sequence of the protein. We have found the N-terminal sequence (GPIVTYYGQN) most homologous to a group of proteins called “Hevamine A-like proteins”. Notably, it has no homology with the MAP30 protein.

Hevamine A-like proteins are one of several members of a family of plant chitinases and lysozymes produced by plants known as “plant defensins” protective against microbial infections [16]. There are reports that establish the antitumor, antioxidant, antiviral, and anti-inflammatory properties of chitinases and chitinous materials [17,18,19,20]. Moreover, chitinases are used in other industries. For example, chitinases are used in the food industry to produce single cell proteins, to make functional foods, and for biocontrol [17]. If MoMo30 is homologous to this group of chitinases, it could potentially be used outside a clinical setting. In addition, Hevamine-A-like proteins are known to be a carbohydrate binding proteins. Other carbohydrate binding proteins such as CyanoVirin-N have been shown to have the ability to bind to the envelope glycoprotein sgp120 on HIV and inhibit viral replication [21,22]. CyanoVirin-N shares a similar motif with MoMo30, suggesting MoMo30 may use a similar mechanism [23]. Carbohydrate binding proteins are gaining popularity as potential antimicrobials. Because of their unique binding specificity, carbohydrate binding proteins are ideal candidates for vaccine development. For example, the snowdrop (*Galanthus nivalis*) lectin (GNA) is a mannose-binding lectin. Its binding specificity is so unique, it only interacts with nonreducing terminal α-D-mannosyl groups [24,25].

We expect MoMo30 to be a chitinase with carbohydrate binding properties. Such proteins are known as chi-lectins, or chitinase-like proteins (CLPs) and are a members of the glycosyl hydrolase 18 (GH18) [26] family of proteins. GH18 proteins are characterized by a (β/α)_8_ barrel folding structure. This structure possesses notable stability [26]. Further research into the homology of MoMo30 with chitinases, chi-lectins, and other carbohydrate binding proteins could provide insight into the inhibitory mechanism of action.

## 5. Conclusions

This study was focused on determining if extracts of *Momordica balsamina* leaves contain anti-HIV-1 activity and to determine the nature of the active ingredient(s) responsible for this activity. Infectivity assays confirmed the presence of anti-HIV-1 activity in extracts of *M. balsamina* leaves. We generated a dose–response curve and showed that *M. balsamina* leaf extract has anti-HIV-1 activity from 20–1000 µg/mL (Figure 1A). At a 20 µg/mL dose, HIV-1_NL4-3_ infection had decreased by approximately 75%. By 200 µg/mL, infection decreased by more than 95%. The data suggest that the water-soluble extract contained activity. The extract showed minimal cellular toxicity at the same concentrations in an M.T.T. assay. These data suggest there is little to no cellular toxicity over the inhibitory range.

The solubility data suggested that the active ingredient in the extracts had differential solubility properties. The ability to dissolve in aqueous and organic environments suggests that the active ingredient is a molecule large enough to possess polar and nonpolar domains (amphipathic). Together with data from the molecular cutoff filters, the results suggested that we were dealing with a large molecule with different domains of various solubility. These data suggested that the active agent of the extract was a large molecule, such as a protein. Since the prominent protein in the SDS-PAGE gel was a 30 kD protein, this confirmed our hypothesis. The 100 kD filter partly retained the antiviral activity. We speculate that the protein involved can form multimers and be retained by the larger filters. Due to its origin and size, we have called the protein MoMo30 for Momordica protein 30 kD.

## Figures and Tables

**Figure 1 ijerph-19-15227-f001:**
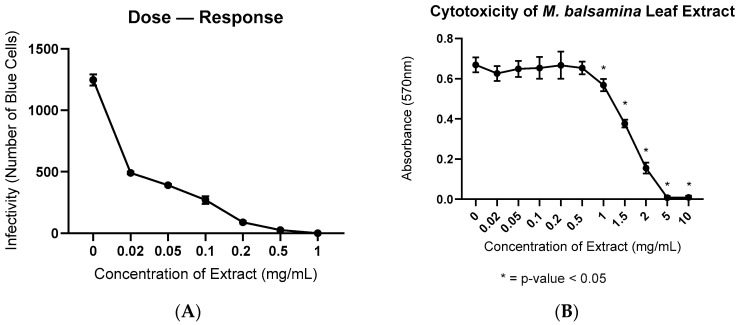
(**A**) This is a dose–response curve measuring the viral inhibition of the *M. balsamina* leaf extract against 1 ng of NL4-3. Treatment concentrations range from 0–1 mg/mL of extract. This assay was done in triplicate with the average infectivity for each concentration shown; (**B**) MTT assay. Concentrations of *M. balsamina* extract tested for cytotoxicity ranged from 0–10 mg/mL. The assay was done in triplicate.

**Figure 2 ijerph-19-15227-f002:**
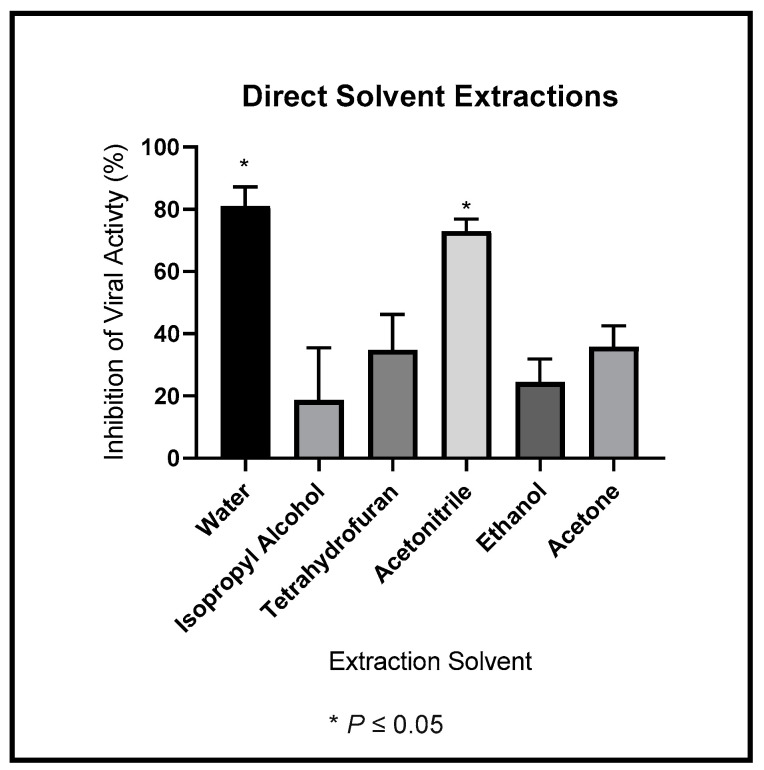
Dried *M. balsamina* leaves were extracted into water, isopropyl alcohol, tetrahydrofuran, acetonitrile, ethanol, and acetone. We used the extracts to treat cells infected with 1 ng of HIV-1_NL4-3_. We show the average antiviral inhibitory effects.

**Figure 3 ijerph-19-15227-f003:**
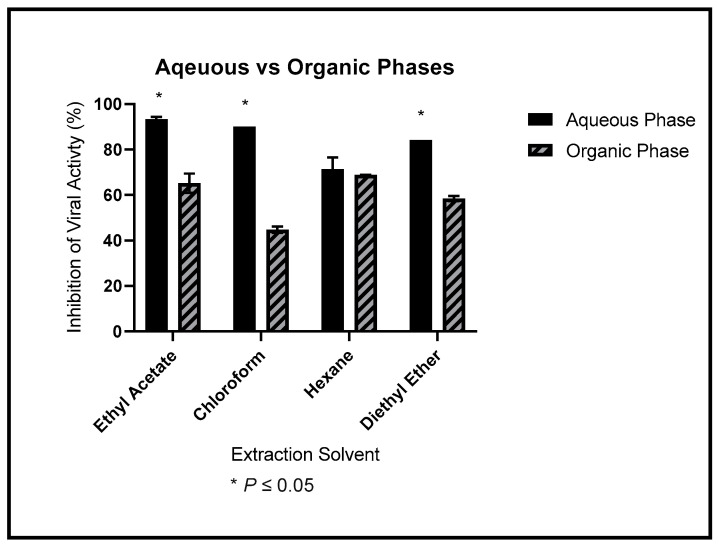
Dried *M. balsamina* leaves were extracted into ethyl acetate, chloroform, hexane, and diethyl ether. We separated the extracts into an aqueous and organic layer and used the layers to treat cells infected with 1 ng of HIV-1_NL4-3_. We show the average anti-HIV-1 inhibitory effects. The solid bar represents the aqueous layer, and the dashed bar represents the organic layer. The assay was done in triplicate.

**Figure 4 ijerph-19-15227-f004:**
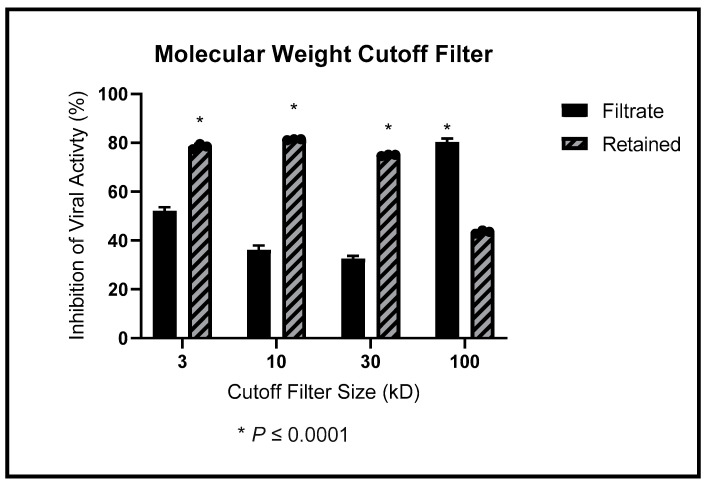
We passed through 3 kD, 10 kD, 30 kD, and 100 kD filters. The filtered and retained liquids were used to treat cells infected with 1 ng of HIV-1_NL4-3_. We show the average anti-HIV-1_NL4-3_ inhibitory effects. Solid bars present the filtrate, and dashed bars represent the retained liquid.

**Figure 5 ijerph-19-15227-f005:**
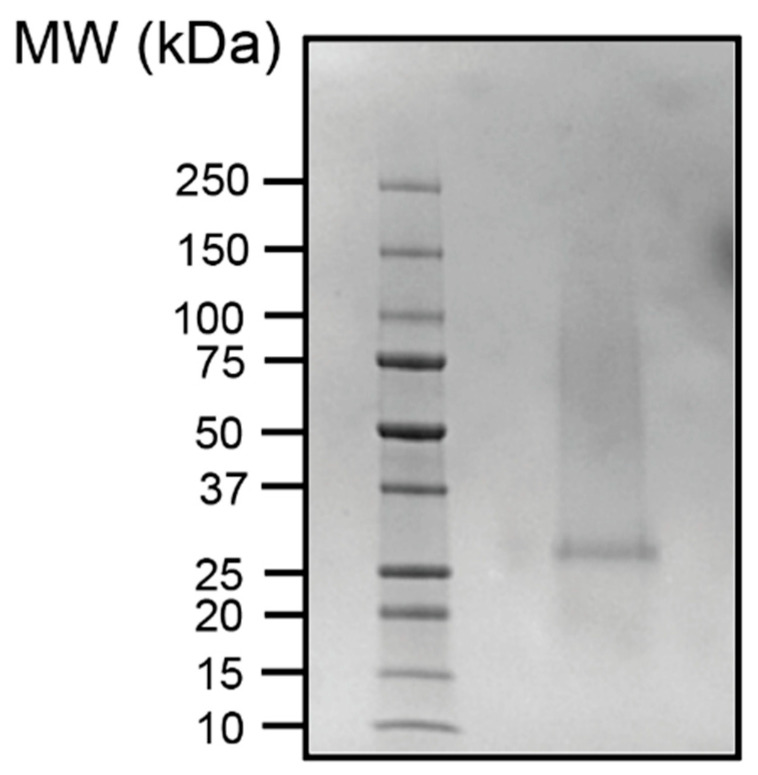
*M. balsamina* leaf extract was run on an SDS-PAGE gel. The gel was stained in C.B.B. to elucidate any possible proteins in the extract. Only one protein band appeared at approximately 30 kD.

**Figure 6 ijerph-19-15227-f006:**
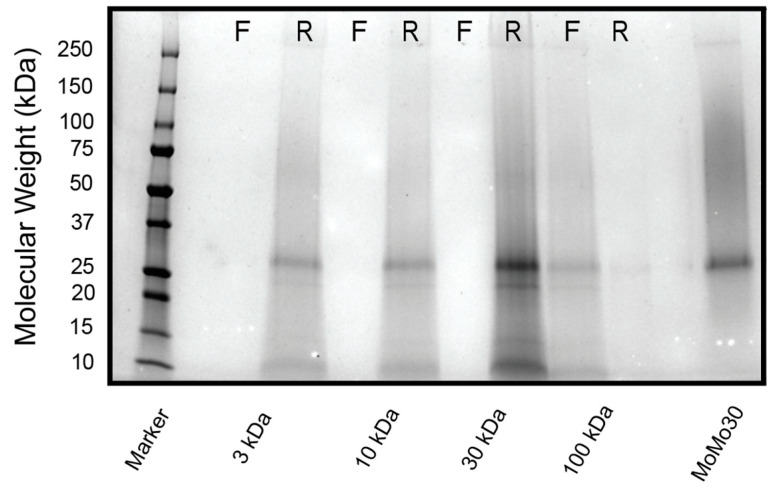
*M. balsamina* leaf extract was run on an SDS-PAGE gel after passing through a series of molecular weight cutoff filters. “R” represents the liquid retained by the filter, and “F” represents the filtrate from the filter. The gel was stained in C.B.B. to elucidate any possible proteins in the extract.

**Table 1 ijerph-19-15227-t001:** Dunnett’s Multiple Comparisons Test.

Concentration of Leaf Extract (mg/mL)	*p*-Value
0 vs. 0.02	0.6643
0 vs. 0.05	0.9913
0 vs. 0.1	0.9973
0 vs. 0.2	>0.9999
0 vs. 0.5	0.9972
0 vs. 1	0.0220 *
0 vs. 1.5	<0.0001 *
0 vs. 2	<0.0001 *
0 vs. 5	<0.0001 *
0 vs. 10	<0.0001 *

* = *p* < 0.05.

**Table 2 ijerph-19-15227-t002:** β-galactosidase Activity.

Volume of Water/Extract (µL)	β-gal Enzyme + Water	β-gal Enzyme + Extract (100 µg/µL stock)
1	101 ± 0.8	102 ± 0.9
5	102 ± 0.9	102 ± 0.2
10	103 ± 0.1	102 ± 0.9
100	97 ± 0.9	98 ± 2.7

**Table 3 ijerph-19-15227-t003:** Tukey’s Multiple Comparisons Test.

Extraction Solvents	*p*-Value
Water vs. Isopropyl Alcohol	<0.0001 *
Water vs. Tetrahydrofuran	0.0008 *
Water vs. Acetonitrile	0.8992
Water vs. Ethanol	0.0001 *
Water vs. Acetone	0.0010 *
Isopropyl Alcohol vs. Tetrahydrofuran	0.3795
Isopropyl Alcohol vs. Acetonitrile	0.0002 *
Isopropyl Alcohol vs. Ethanol	0.9748
Isopropyl Alcohol vs. Acetone	0.3183
Tetrahydrofuran vs. Acetonitrile	0.0043 *
Tetrahydrofuran vs. Ethanol	0.7761
Tetrahydrofuran vs. Acetone	>0.9999
Acetonitrile vs. Ethanol	0.0006 *
Acetonitrile vs. Acetone	0.0054 *
Ethanol vs. Acetone	0.7033

***** = *p* < 0.05.

**Table 4 ijerph-19-15227-t004:** Šídák’s Multiple Comparisons Test.

Aqueous Phase vs. Organic Phase	*p*-Value
Ethyl Acetate	<0.0001 *
Chloroform	<0.0001 *
Hexane	0.7774
Diethyl Ether	<0.0001 *

* = *p* < 0.05.

## Data Availability

Not applicable.

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
