# Peer review of "Identification of a Novel Anti-HIV-1 Protein from Momordica balsamina Leaf Extract"

_ijerph, 2022, doi:10.3390/ijerph192215227_

Round 1
Reviewer 1 Report
- The aim in introduction and abstract, in the title not matched, you should write in the title (extracts from leaves)!!
-Line 22: make M italic (M. balsamina).
-Line 22: modify into (The results showed that the antiviral...)., also delete purpose, methods, results, and discussion words from the abstract, you should write it directly.
- A recent studies relate to antimicrobial power of M. balsamina should be added in the introduction.
-Line 69: (and then added water)?! write the purpose of adding water for cleaning, or extraction?
-Do not make the aim bold in the introduction.
-The mixture was boiled for 30 minutes and dried into a powder? What is exact temperature, I am afraid the high temperature will decrease the bioactivity of the extract? Are you follow the reference for the extraction? Where is the original reference??
-Modify xG into x g. (in line 72).
-For the title (2.2. extraction preparation) make a general preparation of raw materials for all extraction solvents. Also, write sub section related with aqueous extraction only.
-Line 78-79: why you take the water extract to make organic extraction, you should extract directly without using the water extract to make a comparison and see the best and optimum solvent???
- Must write ml as mL apply for similar. all units have problems.
-83-85 un clear totally.
- The authors did not write the model of the instruments, and the reagents sources in the material used and methods.
-Also you did not write the references for many methods??
-Experimental design and methodology incomplete.
- The language level is very bad, as well for standard units.
-line 156 the plant must be italic! apply for all.
-Figure 2. you should specify each abbreviation in the figure under the figure with the complete word between ( ).
- Line 177: Why did not use water extract!
- The conclusion should be after the discussion. (conclusions is very long). the discussion is short and no.
-More numerical results in abstract should be added!
- In fig.5 you did not specify the marker, why did not use various sample with different solvent in SDS PAGE?? to see their profile?
- Figures design is bad written.
- The aim not achieved, since you dont conclude in the abstract regarding the optimum condition and optimum solvent with antiviral activity as well.
- in the abstract where is antiviral power findings??
-line 144 why you used only water solvent?
-The title has concern e.g., protein isolated? are you isolated the protein, and in each pH? plant extract! you should writ leaf extract
Author Response
Thanks for the thorough review and your comments. Point-by-Point responses are attached.

Author Response
Thank you for the thorough review. See point-by-point responses attached.

Round 2
Reviewer 1 Report
- The authors did not address my comments and recommendations sufficiently.
-Although the authors has revised the manuscript, the manuscript still has various concerns nd the language level is very bad.
-A native speaker and expert as well to revise the manuscript for several scientific and language concerns.
- Abstract must be improved with numerical and significant results.
-The author did not uniform the writing style e.g., 100ºC in line 66 and line 4 for 4°C, also Figures why make it sometimes italic? also Figure 1A, and Figure 1a?? so harmonies your writing style.
-In Fig.1: write the captions under the figure (Dose-Response) etc., should be under no above, and repeat it in line 179 before you start in define its meaning. Overall, the titles must be matched in the figures and their captions (for example Fig. 1B MMT...), you must rewrite the titles of all figures in good and understood way!!
-Where is the reference for the sentence started with line 22?
- Why you did not cite the relevant studies e.g., inhibition of HIV-1 Replication by Balsamin, a Ribosome Inactivating Protein of Momordica balsamina (https://www.ncbi.nlm.nih.gov/pmc/articles/PMC3764001/).
-The introduction is incomplete!!!
-The methodology in complete and confusing to repeat in upcoming studies, the reagents sources, the instruments models non written to reproduce this study e.g., lyophilizer, centrifuge, etc.
-I am disappointed that, the authors did not write about the health benefits of the M. balsamina leaf in the introduction.
-Line 61: In section 2.1. write the storage conditions of the fresh leaf.
Lines 63-65: What is the drying conditions of the fresh leaves? in oven? or in sun? what is the time and temperature used?
-line 69: add "at" after centrifuged.
-line 69: no space between x ad g
-Line 156: rewrite the equation in scientific format.
- I confused about the section 2.2. and section 2.5.? Sec.2.2. for aqueous extraction only, so modify the title? What is the aim of this preparation, since you will do aqueous extraction again in sec. 2.5.??
-In addition What is the aim of using different solvents? Antiviral only? Why did not test these extracts based on various solvents on cytotoxic assay and IC50.
-line 187: change ml to mL, Apply that throughout the manuscript.
-lines 217-221: There are various problem in tables design, these are non scientific tables, the title must be above the table, What is the mean of: sennAsteric indicates statistical significance?
-Line 38: What is the mean of ac?
-The tables titles must be revised and rewritten, write the title above the table.
-Line 230: Dried M. balsamina? add leaf after dried... then add were...
-Please explain why you conducted SDS PAGE for water extract only?
-In line 250 its confused? you only work on water, but in line 250 you stated you do for all solvents. thus in line 252, be accurate about the used solvent type? Also modify the title of Fig. 5? which type of solvent??
-Line 273: modify ml to mL
- Write the conclusions after discussion part.
Reviewer 2 Report
Figure 6: Why do black lines separate the gel image? If each MW cutoff was run on a different gel, please provide full gel images with a ladder in each gel.
Author Response
Please see attached comments.

Round 3
Reviewer 1 Report
The following minor comments should be addressed for further consideration.
-Line 24: Add space between 0.02 and mg/mL, apply throughout the manuscript.
-Line 25: write as follow: P ≤ 5, and also in lines 27 and 31, apply throughout the manuscript. Additionally, I confused you wrote P ≤ 5, while in the figures you write only < 5????
-line 27: remove underline.
-Line 81 remove the space between 100 and °C, apply throughout the manuscript.
Line 84: remove the space between 100 and g, apply throughout the manuscript. Note that the only case to remove the space in "%", e.g., 75%.
-Line 81, 86, and 88, complete the instruments model with the manufacturer country.
-Line 146: edit µl to µL, apply throughout the manuscript.
-Line 84: remove the space between 1 and mL, apply throughout the manuscript.
-Line 171: rewrite the equation, percentage of viral inhibition must be came initially before=
-Line 213, I told you a lot regarding move the table title above the table, and the quality of the table must be improved, apply for Tables 3 and 4 as well.
-Discussion section: there is lack of discussion, its seems that the authors reviewed the recent studies, but the discussion must be from the own understanding according to the current findings and the scientific law. What is the novelty of the current findings? no only write from the recent studies.
-line 327: merge the future directions with the conclusions (section 5) in one title.
-Check the references bibliography according to IJERPH format.
